# Borderline Personality Features and Mate Retention Behaviors: The Mediating Roles of Suspicious and Reactive Jealousy

**Virgil Zeigler-Hill ***[ID] **and Jennifer Vonk** [ID]

Department of Psychology, Oakland University, Rochester, MI 48309, USA; vonk@oakland.edu
* Correspondence: zeiglerh@oakland.edu

**Abstract:** We investigated the roles that suspicious jealousy and reactive jealousy might play in the associations between borderline personality features (BPF) and mate retention behaviors. Study 1 ($N$ = 406) found that BPF had positive indirect associations with benefit-provisioning behaviors and cost-inflicting behaviors through suspicious jealousy but not through reactive jealousy. Study 2 ($N$ = 334 (a dyadic sample of 167 romantic couples)) revealed actor effects such that BPF had positive indirect associations with benefit-provisioning behaviors and cost-inflicting behaviors through suspicious jealousy for both men and women. In addition, the positive association between BPF and benefit-provisioning behaviors was mediated by reactive jealousy in women but not in men. The only partner effect that emerged from these analyses showed that BPF in women were negatively associated with the benefit-provisioning behaviors reported by their male partners. Discussion focuses on the implications of these results for the function that jealousy might serve in the strategies used by individuals with BPF to maintain their romantic relationships.

**Keywords:** borderline; personality; jealousy; romantic relationships; mate retention





## 1. Introduction

Borderline personality disorder (BPD) is a psychological disorder characterized by instability. This instability manifests in many areas for those with BPD, including their affective experiences, views of themselves, and relationships with others [1]. Many of the diagnostic criteria for BPD are either clearly interpersonal (e.g., frantic attempts to avoid potential rejection or abandonment) or tend to emerge in response to negative interpersonal events (e.g., engagement in self-harm following interpersonal conflicts). The interpersonal difficulties that characterize those with BPD have been shown to extend to their romantic relationships [2–5]. In fact, the pattern of intense and unstable romantic relationships may be one of the most useful criteria for diagnosing BPD [6]. This research was intended to extend previous findings by examining the connections that borderline personality features (BPF) have with the strategies used to maintain romantic relationships as well as what role—if any—suspicious and reactive jealousy may play in these associations.

### 1.1. Borderline Personality Features

As with other personality disorders, the current view of BPD is based on a categorical model [1]. However, there is, at best, limited empirical support for this categorical conceptualization of BPD [7,8], which aligns with the argument that the thresholds used to diagnose many of the personality disorders are at least somewhat arbitrary [9]. This discussion has led to various arguments in favor of the adoption of a dimensional conceptualization of BPD that would recognize meaningful individual differences in severity beyond what can be captured by a categorical diagnosis [10,11]. This perspective also aligns with the view that there may be subclinical levels of BPD symptoms in the general population that may be linked with interpersonal functioning and psychological well-being despite not meeting

the clinical threshold for BPD [12]. It is these individual differences in traits occurring at subclinical levels that we refer to as BPF.

One approach to understanding the dimensional nature of personality disorders has been to use the Five Factor Model of basic personality [9,13]. There is clear evidence that each of the personality disorders can be conceptualized as problematic combinations of these basic personality traits [14,15]. This perspective has been applied to BPD such that it is viewed as primarily consisting of neuroticism with some elements of agreeableness, conscientiousness, and openness to experience [14,16], leading to the development of the Five Factor Borderline Inventory (FFBI) [12] and the FFBI-short form [17] to assess subclinical levels of BPF.

Individuals high in BPF engage in various detrimental behaviors including non-suicidal self-injurious behaviors [18], misuse of substances [19], impulsive sexual behaviors [20], and physical aggression [21]. The potentially self-destructive behaviors that characterize those with high levels of BPF are believed to serve as problematic means to regulate their negative emotional states [22]. Given the nature of these self-destructive behaviors, individuals with high levels of BPF tend to experience lower quality of life and those close to them often feel burdened by their destructive behaviors [23].

### 1.2. Borderline Personality Features and Romantic Relationships

There is little doubt that romantic relationships are impacted by the interpersonal issues that characterize BPF. For example, individuals with elevated levels of BPF—and their romantic partners—report a range of aversive outcomes including less satisfaction, greater conflict, greater stress, more break-ups, and more intimate partner violence compared to those low in BPF [24,25]. Individuals with high levels of BPF tend to experience difficulties in balancing issues concerning intimacy and autonomy, which often leads to perceptions of rejection and unmet needs, along with feelings of mistrust, hostility, and anger [5,25–27]. In addition, individuals with high levels of BPF often display intrusive, vindictive, and domineering behaviors in their relationships [26] and are often unaware of the negative impacts that their behaviors have on those around them [28]. These issues converge to create a situation in which the relationships of individuals high in BPF often involve an intense need for closeness and attention accompanied by feelings of rejection, distress, and hostility [29].

A pattern of conduct indicative of frantic attempts to prevent rejection or abandonment is one of the diagnostic criteria for BPD [1]. These actions could include mate retention behaviors (MRB), which are strategies that an individual can use to make it less likely that a partner will break up with them or engage in infidelity [30,31]. MRB can be divided into two broad types [32]. Benefit-provisioning behaviors are relatively innocuous acts that emphasize the advantages associated with the relationship to the partner and provide the partner with incentives to maintain the relationship (e.g., complimenting the partner). In contrast, cost-inflicting behaviors are relatively harsh strategies that involve imposing—or threatening to impose—unpleasant consequences on the partner if they were to dissolve the relationship or be unfaithful (e.g., threatening to cause harm to the partner if they appear to be flirting with someone else). MRB are characteristic of human relationships given the long-term adaptive advantage of maintaining monogamous bonds for child-rearing, but excessive use of these strategies can reduce relationship functioning and may even contribute to extreme partner violence [33].

Previous studies show that people high in BPF exhibit greater MRB [34,35]. More specifically, there is often a positive association between BPF and cost-inflicting behaviors, whereas the association that BPF have with benefit-provisioning behaviors tend to be weaker and less reliable. These results are consistent with findings that high levels of BPF are associated with the use of instrumental aggression in response to perceived rejection [36]. Because individuals with high levels of BPF have difficulty regulating negative emotions—including jealousy—they may also experience reduced empathy toward their partners, which may, in turn, encourage the use of cost-inflicting strategies that neg-

atively impact their romantic relationships [34]. Although cost-inflicting behaviors can be effective in some situations, they are inherently risky strategies because their aversive nature may unintentionally prompt romantic partners to either end the relationship or engage in infidelity, which may perpetuate this destructive cycle.

### 1.3. The Importance of Trust

Individuals high in BPF often find it difficult to understand what others are thinking or feeling [37,38], and they consistently make more negative attributions with regard to the trustworthiness of other people [39–44]. This pattern includes the tendency to make malevolent attributions for ambiguous or neutral behaviors [45,46]. These attributional biases likely contribute to a lack of trust in close friends and romantic partners [47]. Indeed, recent advances in the conceptualization of BPD suggest that issues surrounding trust serve as the fundamental core of this disorder [48–50], which is consistent with the finding that BPD involves high levels of antagonism (i.e., a basic personality trait characterized by trust issues) [51].

Trust is an essential quality that allows individuals to experience healthy and satisfying intimate relationships [52]. A general distrust in others may lead individuals high in BPF to act preemptively in an effort to avoid potential threats [34]. For example, individuals high in BPF may exhibit aversive behaviors when they fear being rejected or abandoned, which are issues that tend to be prominent in their thinking [5,35]. However, these aversive behaviors—including cost-inflicting MRB—may paradoxically endanger the relationship that the individual is attempting to save. For example, hypersensitivity to rejection may lead to increased criticizing, which, in turn, may lead the partner to withdraw and perpetuate the cycle [24].

### 1.4. Borderline Personality Features and Jealousy

BPF have been shown to be associated with jealousy in romantic relationships [21,29,53,54]. These characteristic feelings of jealousy may be due to preoccupation with rejection and abandonment, which may lead individuals high in BPF to experience interpretive biases regarding the behaviors and intentions of their romantic partners (e.g., suspect that they may be engaging in infidelity) [29,53]. In essence, the jealousy that is experienced by individuals high in BPF may be due to their lack of trust in others. The intense desire for close relationships combined with concerns about rejection and abandonment—stemming from their lack of trust in others—are likely the reasons that individuals high in BPF tend to report disorganized attachment styles [55].

Although high levels of BPF have been shown to be associated with jealousy, the story is complicated by the distinction between reactive jealousy and suspicious jealousy [56]. Reactive jealousy is evoked in the face of undeniable evidence that one has been betrayed, such as in the case of witnessing one's partner engaged in sexual behavior with someone else. In this circumstance, it would be completely reasonable for the person to experience jealousy. In contrast, suspicious jealousy arises in the absence of any clear evidence of betrayal and is predicted by lower levels of security and self-esteem. This form of jealousy is problematic because it can promote unnecessary stress and negative emotional states. It is important to note that jealousy cannot prevent betrayal on its own. However, jealousy can promote the use of MRB, which may help to protect the individual from the considerable costs of relationship dissolution or infidelity. Given the increased likelihood of perceiving others' behavior as threatening (even when it is not), and given the intense fear of abandonment as well as the heightened reactivity of individuals with BPF in response to perceived threats, it seems reasonable to expect increased levels of suspicious jealousy for individuals high in BPF. Because individuals high in BPF fear abandonment and are willing to engage in extreme, even self-destructive behaviors to maintain their romantic relationships, we would expect heightened feelings of suspicious jealousy to promote both benefit-provisioning and cost-inflicting MRB in these individuals. Thus, we predict that BPF will have positive associations with both cost-inflicting and benefit-provisioning behav-

iors that will be mediated by suspicious jealousy. We were less certain about how reactive jealousy would be associated with BPF and MRB. The reason for our uncertainty was that individuals should exhibit an emotional response to clear signs of infidelity regardless of whether they have high levels of BPF.

Although we anticipated that suspicious jealousy would mediate the associations that BPF had with benefit-provisioning and cost-inflicting behaviors, it is important to emphasize that we cannot address issues concerning causation with the correlational data from the present studies. We fully acknowledge that other models could potentially be used to understand the associations between BPF, jealousy, and MRB. However, we think that the proposed model suggesting that BPF have indirect associations with MRB through suspicious jealousy provides a plausible lens for understanding one of the mechanisms that may help explain the connections between BPF and MRB.

There are gender differences in BPD such that women are far more likely than men to receive this diagnosis (i.e., 75% of individuals who receive a BPD diagnosis are female) [1]. In addition, gender has been observed to influence the relationships between BPF and certain outcomes. For example, BPF are positively associated with the use of the mate retention tactic of emotional manipulation in men but not women [34]. In contrast, BPF are positively associated with certain forms of intimate partner aggression (e.g., verbal aggression) for women but not men [57]. As a result, we decided to conduct exploratory analyses to investigate whether the associations that BPF had with mate retention behaviors through suspicious or reactive jealousy differed between men and women, even though we did not have predictions concerning gender.

## 2. Study 1: Monadic Sample

In Study 1, we assessed whether the associations that BPF had with benefit-provisioning and cost-inflicting MRB were mediated by suspicious jealousy and reactive jealousy.

### 2.1. Materials and Methods

The participants comprised 453 community members who were recruited through Prolific and paid $15 for their participation. The only requirement for participating was that individuals had to be involved in a heterosexual romantic relationship for at least six months. We excluded data for 47 participants due to issues that indicated inattentive responding (e.g., large amount of missing data, short completion time). The average relationship length for the final 406 participants (209 women, 197 men) was 12.39 years (SD = 12.06 [range = 6 months–54 years); 79% were married, 6% were cohabiting, 5% were engaged, and 10% were dating. The average age for the final participants was 43.09 years (SD = 13.07; range = 18–76 years). This study was conducted in accordance with the Declaration of Helsinki, and approved by the Institutional Review Board of Oakland University (IRB Protocol #IRB-FY2022-380 approved on June 15, 2022).

Borderline personality features. We assessed BPF using the Five Factor Borderline Inventory-Short Form (FFBI-SF) [17]. The FFBI-SF (48 items; e.g., "I can be so different with different people that I wonder who I am" [$\alpha$ = 0.96]) is a shortened version of the original FFBI [12], which measures BPF from the framework of the Five Factor Model of personality. We decided to focus on the overall composite score for the FFBI-SF.

Jealousy. We used the Multidimensional Jealousy Scale [56,58] to assess suspicious jealousy (16 items; e.g., "I suspect that [my partner] is secretly seeing someone of the opposite sex" [$\alpha$ = 0.91]) and reactive jealousy (8 items; e.g., "[My partner] shows a great deal of interest or excitement in talking to someone of the opposite sex" [$\alpha$ = 0.92]).

Mate Retention Behaviors. We measured MRB with the Mate Retention Inventory-Short Form (MRI-SF) [59]. The MRI-SF assesses two forms of mate retention: benefit-provisioning behaviors (16 items; e.g., "Complimented my partner on their appearance" [$\alpha$ = 0.88]) and cost-inflicting behaviors (22 items; "Got my friends to beat up someone who was interested in my partner" [$\alpha$ = 0.86]).

### 2.2. Data Analysis

We conducted parallel multiple mediation analyses to examine whether suspicious jealousy and/or reactive jealousy would mediate the associations that BPF had with benefit-provisioning behaviors and/or cost-inflicting behaviors. We used the PROCESS macro for these analyses [60]. It is important to note that although we used a mediation model to conceptualize the connection between these variables, we do not intend to infer causality because we are dealing with cross-sectional data that is correlational in nature.

### 2.3. Results

Table 1 shows the descriptive statistics and zero-order correlations. BPF had very large positive correlations with suspicious jealousy and cost-inflicting MRB. Suspicious jealousy had small positive correlations with reactive jealousy and benefit-provisioning behaviors as well as a very large positive correlation with cost-inflicting behaviors. Benefit-provisioning behaviors had a very large positive correlation with cost-inflicting behaviors.

**Table 1.** Study 1: Intercorrelations and Descriptive Statistics.

|  | **1** | **2** | **3** | **4** | **5** |
|---|---|---|---|---|---|
| 1. Borderline Personality | — | | | | |
| 2. Suspicious Jealousy | 0.54 ** | — | | | |
| 3. Reactive Jealousy | 0.06 | 0.13 * | — | | |
| 4. Benefit-Provisioning | 0.01 | 0.11 * | −0.09 | — | |
| 5. Cost-Inflicting | 0.41 ** | 0.58 ** | −0.06 | 0.43 ** | — |
| Mean | 1.78 | 1.60 | 4.62 | 1.44 | 0.40 |
| Standard Deviation | 0.64 | 0.75 | 1.45 | 0.54 | 0.36 |

* $p < 0.05$; ** $p < 0.001$.

#### 2.3.1. Parallel Multiple Mediation

The results of the parallel multiple mediation analyses appear in Figure 1. These analyses revealed the expected positive association between BPF and suspicious jealousy, whereas there was no association between BPF and reactive jealousy. BPF had the expected positive association with cost-inflicting behaviors, whereas it was not associated with benefit-provisioning behaviors. Tests of mediation revealed that BPF had indirect associations with benefit-provisioning behaviors and cost-inflicting behaviors through suspicious jealousy but not through reactive jealousy.

#### 2.3.2. Moderated Parallel Multiple Mediation

Although we did not have predictions concerning gender, we conducted exploratory analyses to examine whether gender moderated the indirect associations that BPF had with benefit-provisioning behaviors and cost-inflicting behaviors through suspicious jealousy. These analyses revealed that gender moderated the association that BPF had with benefit-provisioning behaviors ($\beta = -0.12$, $t = -2.37$, $p = 0.018$, $CI_{95\%}$ [−0.21, −0.02]). Simple slopes tests revealed a negative association between BPF and benefit-provisioning behaviors for men ($\beta = -0.19$, $t = -2.44$, $p = 0.015$, $CI_{95\%}$ [−0.34, −0.04]) but not for women ($\beta = 0.04$, $t = 0.57$, $p = 0.568$, $CI_{95\%}$ [−0.10, 0.19]). However, gender did not moderate the associations that BPF had with suspicious jealousy, reactive jealousy, or cost-inflicting behaviors. Additionally, there was no support for moderated mediation because the magnitude of the indirect associations that BPF had with benefit-provisioning behaviors and cost-inflicting behaviors through suspicious jealousy did not differ between men and women.

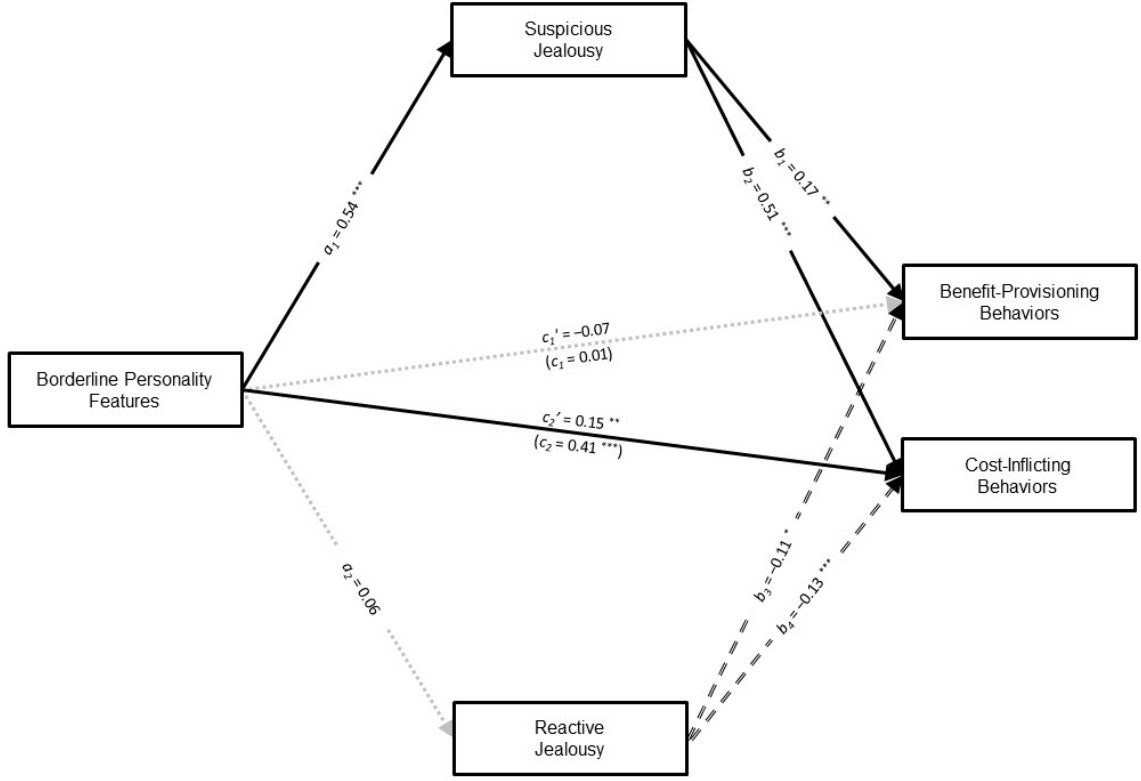

**Figure 1.** The results of the parallel multiple mediation analysis with suspicious jealousy and reactive jealousy mediating the associations that borderline personality features had with benefit-provisioning behaviors and cost-inflicting behaviors. Note: The positive associations are indicated by solid black arrows, the negative associations are indicated by dashed black arrows, and the nonsignificant associations are indicated by dotted grey lines. * $p < 0.05$; ** $p < 0.01$; *** $p < 0.001$.

*2.4. Discussion*

As expected, BPF had positive indirect associations with both benefit-provisioning and cost-inflicting MRB through suspicious jealousy. However, there was no significant direct relationship between BPF and benefit-provisioning behavior. That is, individuals high in BPF were not significantly more likely than those low in BPF to engage in benefit-provisioning MRB. Rather, the association that did exist between BPF and benefit-provisioning behavior was due to those with higher BPF expressing more suspicious jealousy, which, in turn, was associated with benefit-provisioning behaviors. When we examined whether gender moderated the associations that BPF had with MRB, we found that BPF was negatively associated with benefit-provisioning behavior for men but not for women. However, there were no effects of moderated mediation. These results indicate that suspicious jealousy—but not reactive jealousy—influences the connections that BPF have with benefit-provisioning and cost-inflicting MRB. This pattern suggests that individuals high in BPF are more likely to be suspicious of their partner's behavior even in the absence of overt cues of infidelity or other betrayals. In turn, this suspicious jealousy may then lead to both benefit-provisioning and cost-inflicting behaviors to maintain the relationship.

**3. Study 2: Dyadic Sample**

We examined BPF in romantic couples where the experience of suspicious and reactive jealousy and engagement in MRB could be examined in both members for the first time. This allowed us to examine both actor effects and partner effects to better understand the connections between BPF, jealousy, and MRB in both members of romantic dyads. We adopted the same basic conceptual model in which we explored whether the associations that BPF had with MRB were mediated by suspicious and reactive jealousy, but our dyadic

design also allowed us to examine whether a partner's BPF and/or jealousy predicted the MRB of their partner for exploratory purposes.

### 3.1. Materials and Methods

The participants comprised 394 community members (i.e., 197 romantic couples) who were recruited through Prolific and were paid $15 each for their participation. The only requirement for participating was that individuals had to be involved in a heterosexual romantic relationship for at least six months and their partner also had to be willing to participate. We excluded data for 30 couples for issues that indicated inattentive responding by at least one of the participants in the dyad using the same criteria as Study 1 (e.g., large amount of missing data, invariant response patterns across items). The average relationship length for the final 167 couples was 11.75 years (SD = 10.07; range = 6 months–48 years); 73% were married, 15% were cohabiting, 2% were engaged, and 10% were dating. The average age for women was 39.45 years (SD = 11.1; range = 18–73 years) and the mean age for men was 41.61 years (SD = 11.19; range = 18–84 years).

Borderline personality features. We assessed BPF using the FFBI-SF ($\alpha$ Women = 0.96, $\alpha$ Men = 0.96) as in Study 1.

Jealousy. Suspicious jealousy ($\alpha$ Women = 0.89, $\alpha$ Men = 0.90) and reactive jealousy ($\alpha$ Women = 0.94, $\alpha$ Men = 0.94) were measured using the Multidimensional Jealousy Scale as in Study 1.

Mate Retention Behaviors. Benefit-provisioning behaviors ($\alpha$ Women = 0.87, $\alpha$ Men = 0.86) and cost-inflicting behaviors ($\alpha$ Women = 0.85, $\alpha$ Men = 0.81) were measured using the Mate Retention Inventory-Short Form as in Study 1.

### 3.2. Data Analysis

We analyzed these data using an Actor–Partner Interdependence Mediation Model (APIMeM) [61]. APIMeM is often used for examining indirect effects in dyads because it is able to account for both members of a romantic couple influencing each other. For example, a woman's BPF may be associated with her use of MRB (an actor effect) as well as the MRB used by her romantic partner (a partner effect). We used the MEDYAD macro to conduct these APIMeM analyses [62]. We treated these couples as distinguishable dyads such that the gender of each participant was used to distinguish between the members of the dyad. Although we did not have predictions about gender, treating these couples as distinguishable dyads allowed us to explore the similarities and differences between the actor and partner effects that emerged for men and women.

### 3.3. Results

Table 2 shows the zero-order correlations and descriptive statistics. BPF had large-to-very large positive correlations with suspicious jealousy and cost-inflicting MRB for both women and men. In addition, BPF had a small positive correlation with reactive jealousy and a small negative correlation with benefit-provisioning behaviors for women, whereas these correlations were not significant for men. Suspicious jealousy had a very large positive correlation with cost-inflicting behaviors for both women and men. Suspicious jealousy also had a medium positive correlation with benefit-provisioning behaviors for women but not for men. Reactive jealousy had a medium positive correlation with benefit-provisioning behavior for women but not for men. Benefit-provisioning behaviors had a large-to-very large positive correlation with cost-inflicting behaviors for both women and men. Of note, there were large-to-very large positive correlations for each variable between the members of the romantic dyads (e.g., the BPF for women had a very large positive correlation with the BPF of their male partners). The results of the APIMeM analysis for benefit-provisioning behaviors appear in Figure 2 and the results for cost-inflicting behaviors appear in Figure 3.

**Table 2.** Study 2: Intercorrelations and Descriptive Statistics.

| | **1** | **2** | **3** | **4** | **5** | **6** | **7** | **8** | **9** | **10** |
|---|---|---|---|---|---|---|---|---|---|---|
| 1. Female Borderline Personality | — | | | | | | | | | |
| 2. Female Suspicious Jealousy | 0.36 *** | — | | | | | | | | |
| 3. Female Reactive Jealousy | 0.16 * | 0.13 | — | | | | | | | |
| 4. Female Benefit-Provisioning | −0.17 * | 0.27 *** | 0.20 * | — | | | | | | |
| 5. Female Cost-Inflicting | 0.35 *** | 0.66 *** | 0.07 | 0.37 *** | — | | | | | |
| 6. Male Borderline Personality | 0.40 *** | 0.21 ** | −0.04 | −0.05 | 0.20 ** | — | | | | |
| 7. Male Suspicious Jealousy | 0.24 ** | 0.42 *** | −0.08 | 0.00 | 0.29 *** | 0.42 *** | — | | | |
| 8. Male Reactive Jealousy | 0.09 | 0.01 | 0.35 *** | 0.11 | 0.06 | 0.10 | 0.08 | — | | |
| 9. Male Benefit-Provisioning | −0.19 * | 0.00 | 0.09 | 0.43 *** | 0.09 | −0.03 | 0.12 | 0.06 | — | |
| 10. Male Cost-Inflicting | 0.19 * | 0.31 *** | −0.03 | −0.04 | 0.35 *** | 0.46 *** | 0.60 *** | 0.01 | 0.29 *** | — |
| Mean | 1.94 | 1.69 | 4.77 | 1.40 | 0.47 | 1.82 | 1.57 | 4.69 | 1.41 | 0.40 |
| Standard Deviation | 0.67 | 0.73 | 1.63 | 0.54 | 0.36 | 0.63 | 0.68 | 1.54 | 0.53 | 0.30 |

$* \ p < 0.05;\ ** \ p < 0.01;\ *** \ p < 0.001.$

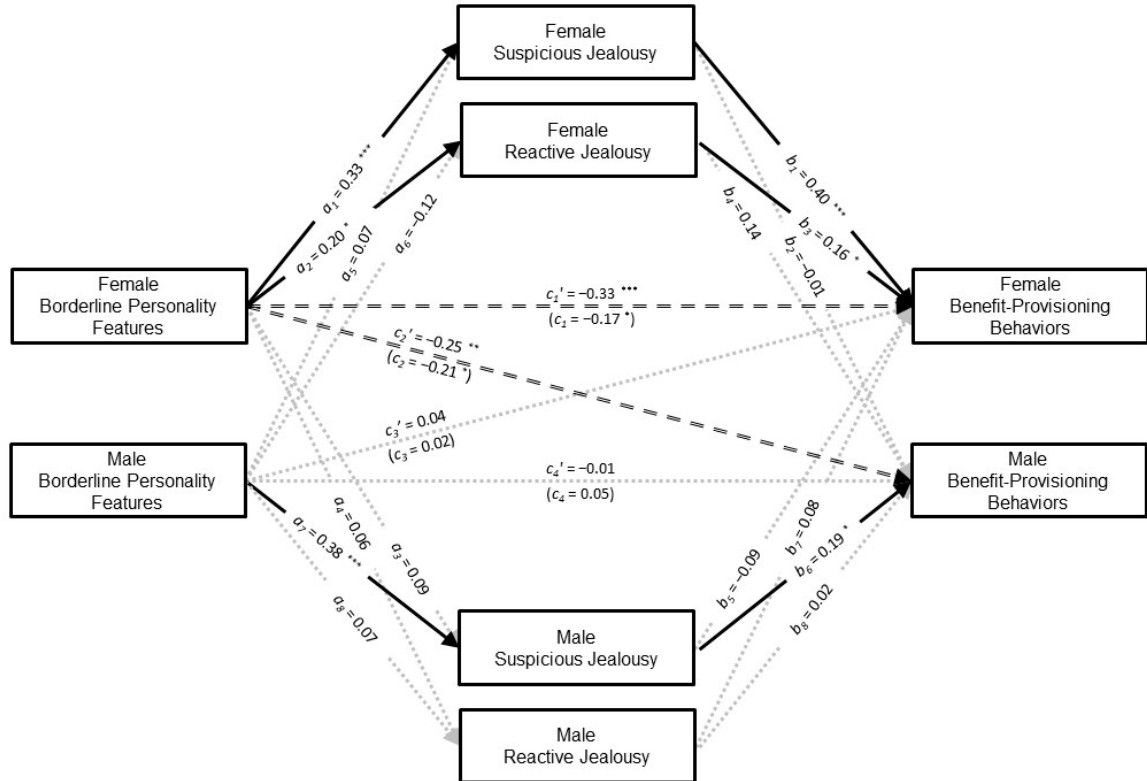

**Figure 2.** The results of the APIMeM analysis with suspicious jealousy and reactive jealousy mediating the associations that borderline personality features had with benefit-provisioning behaviors. Note: The positive associations are indicated by solid black arrows, the negative associations are indicated by dashed black arrows, and the nonsignificant associations are indicated by dotted grey lines. $* \ p < 0.05;\ ** \ p < 0.01;\ *** \ p < 0.001.$

### 3.3.1. Actor Effects for Women

These analyses revealed the expected actor effect for women such that there was a positive association between BPF and suspicious jealousy. In addition, there was also an actor effect for women such that BPF had a positive association with reactive jealousy. BPF had the expected positive association with cost-inflicting behaviors as well as an unexpected negative association with benefit-provisioning behaviors. Tests of mediation revealed that BPF had positive indirect associations with benefit-provisioning behaviors and cost-inflicting behaviors through suspicious jealousy. BPF also had an unexpected positive indirect association with benefit-provisioning behavior through reactive

jealousy, but they did not have an indirect association with cost-inflicting behaviors through reactive jealousy.

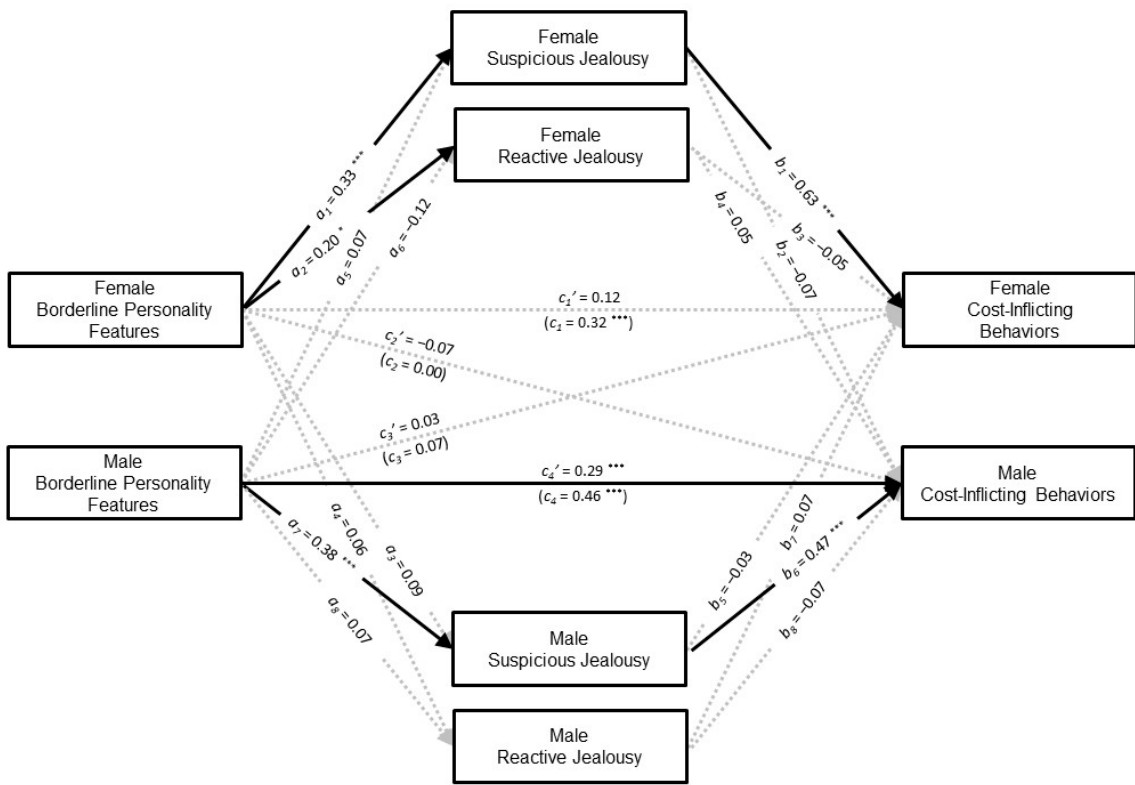

**Figure 3.** The results of the APIMeM analysis with suspicious jealousy and reactive jealousy mediating the associations that borderline personality features had with cost-inflicting behaviors. Note: The positive associations are indicated by solid black arrows and the nonsignificant associations are indicated by dotted grey lines. * $p < 0.05$; *** $p < 0.001$.

### 3.3.2. Actor Effects for Men

The expected actor effect for men emerged such that there was a positive association between BPF and suspicious jealousy. There was no association between BPF and reactive jealousy for men. BPF had the expected positive association with cost-inflicting behaviors and tests of mediation revealed that suspicious jealousy mediated the association that BPF had with cost-inflicting behaviors. Although BPF did not have total or direct associations with benefit-provisioning behaviors, they had a positive indirect association with benefit-provisioning behaviors through suspicious jealousy.

### 3.3.3. Partner Effects for Women

These analyses revealed a partner effect such that the BPF of women was negatively associated with the benefit-provisioning behaviors of their male partners. However, the BPF of women were not associated with the levels of suspicious jealousy or reactive jealousy reported by their male partners. Further, the BPF of women did not have indirect associations with the benefit-provisioning or cost-inflicting behaviors of their male partners through their own jealousy or the jealousy reported by their male partners.

### 3.3.4. Partner Effects for Men

These analyses found no partner effects for men. That is, the BPF of men did not have direct associations with benefit-provisioning or cost-inflicting behaviors of their female partners, nor did they have indirect associations with the MRB of their female partners through their own jealousy or the jealousy reported by their female partners.

*3.4. Discussion*

The results of Study 2 largely mirrored the results of Study 1. That is, the associations that BPF had with benefit-provisioning and cost-inflicting behaviors were mediated through suspicious jealousy for both men and women. This aligns with our prediction that individuals high in BPF may engage in both types of MRB because they would be willing to do whatever was needed to maintain their connections with their romantic partners. However, notably, the pattern of results for cost-inflicting behaviors appeared to be more robust across the studies, whereas the association between BPF and benefit-provisioning behaviors appeared to be somewhat weaker and more complex. Taken together, these results suggest that those high in BPF may more readily resort to cost-inflicting behaviors to preserve their relationships when they are afraid of being rejected or abandoned by their partner. This pattern is consistent with the broad array of interpersonal problems faced by individuals with high levels of BPF.

The results of Study 2 also revealed additional associations for BPF in women that were not observed in Study 1. For instance, the BPF of women were negatively associated with their own benefit-provisioning behaviors as well as the benefit-provisioning behaviors reported by their male partners, whereas similar associations did not emerge for the BPF of men. This suggests that high levels of BPF in women—but not in men—may discourage the use of benefit-provisioning behaviors to maintain relationships. This may be detrimental to relationship functioning (e.g., relationship satisfaction, commitment) because benefit-provisioning behaviors emphasize the advantages for the partner if they continue the relationship. This aversion to engaging in benefit-provisioning behavior may be due to the negative affect that characterizes those with high levels of BPF [63] or their tendency to criticize others when they feel threatened [24]. However, it is unclear why BPF in women seemed to be particularly problematic for the use of benefit-provisioning behaviors in Study 2 since the results of Study 1 showed that there was a negative association between BPF and benefit-provisioning behaviors for men but not for women. These contradictory results suggest that it is important for future studies to attempt to shed additional light on what role—if any—gender plays in the connections that BPF have with romantic outcomes, including MRB.

## 4. General Discussion

The results from a monadic study (Study 1) and a dyadic study (Study 2) provided consistent support for our hypothesis that BPF would have positive indirect associations with benefit-provisioning and cost-inflicting behaviors through suspicious jealousy. That is, our results are consistent with the possibility that BPF is associated with suspicious jealousy, which, in turn, is associated with benefit-provisioning and cost-inflicting MRB. The results for cost-inflicting behaviors suggest that the strategies that individuals high in BPF use to maintain their romantic relationships align with the strategies they use to navigate their other social relationships. That is, individuals with high levels of BPF seem to use intimidation, coercion, and physical force, among other aversive tactics, to prevent partners from terminating the relationship or being unfaithful [34,35]. This reliance on aversive interpersonal tactics for maintaining romantic relationships aligns with the generally hostile approach to interpersonal interactions that has been shown to characterize BPF [64,65].

The results from Study 2 suggest that the use of cost-inflicting MRB—and the suspicious jealousy that may promote these behaviors—do not occur solely as a response to the partner's similar behaviors. There was a lack of observed partner effects despite the fact that individuals with BPF tended to partner with individuals who also exhibit BPF, as demonstrated here and elsewhere [24,64]. However, there is some evidence that stable relationships may help mitigate some of the more problematic behavioral patterns demonstrated by those with BPF [25], so it is possible that we may have observed additional partner effects if we had measured outcomes beyond MRB (e.g., strategies used to navigate conflict). Given that men who had partners with high levels of BPF engaged in fewer

benefit-provisioning behaviors, it appears that some partners may not be mitigating the harmful behaviors of individuals with BPF because they may be upset by the aversive behaviors of their female partners. If men engage in fewer benefit-provisioning behaviors, this may suggest that they engage in fewer behaviors of other types that would provide needed reassurance to their partners with elevated levels of BPF, thus facilitating the perpetuation of the negative relationship cycle. However, our data also seem to suggest that the partners of individuals with BPF are at least not reinforcing their behaviors with their own MRB.

It is important to note that we conducted preliminary analyses for both studies that included the length of the romantic relationships. Although relationship length was associated with many of the other variables (e.g., it had a small negative correlation with suspicious jealousy), it did not moderate any of the associations that we have reported nor did its inclusion in these models substantially alter the reported results. As a consequence, we did not include relationship length in our final analyses.

We predicted that individuals high in BPF may engage in more benefit-provisioning behaviors because they would be willing to engage in any behaviors that might prevent partner abandonment or infidelity, even if the behaviors were costly to themselves [34]. Although benefit-provisioning MRB were not directly associated with BPF in Study 1, they were negatively associated with BPF only for men in Study 1 and only for women in Study 2. These findings, although not predicted, are important because it has been suggested that redirecting those with high levels of BPF to engage in more benefit-provisioning and less cost-inflicting strategies may be a useful treatment strategy [34]. In both studies, BPF had positive indirect associations with benefit-provisioning behaviors through suspicious jealousy. In addition, BPF had a positive indirect association with benefit-provisioning behaviors through reactive jealousy for women in Study 2. We had not predicted this association because we took the perspective that reactive jealousy, which occurs in response to a direct and incontrovertible demonstration of betrayal, such as clear evidence of infidelity, would be beneficial within the context of a romantic relationship and therefore should not be specific to individuals high in BPF.

Although the results were somewhat more complex than anticipated, they aligned with our prediction that individuals high in BPF would be likely to act in ways to maintain their romantic relationships even if those actions were aversive and potentially costly to both themselves and their partners. This prediction was based on a pattern of traits and behaviors characteristic of those high in BPF, such as their intense fear of abandonment [66], insecure attachment [55], mistrust of partners [43], and willingness to engage in self-destructive behaviors [22]. It is unfortunate that the use of aversive behaviors to avoid abandonment may often result in the very outcomes that individuals with BPF are so desperate to prevent and may inadvertently perpetuate a self-defeating cycle.

The present studies had several strengths (e.g., large samples, use of romantic dyads), but there were also important limitations. The first limitation was that the correlational nature of these studies prevents us from drawing causal inferences regarding the connections between BPF, jealousy, and MRB. It is possible that there are alternative causal explanations for the associations that emerged from these studies. For instance, it could be that the chronic experience of suspicious jealousy may actually promote the development of BPF rather than suspicious jealousy resulting from BPF. The second limitation was that we focused exclusively on individuals from the United States in heterosexual relationships, which may limit the generalizability of these results. For example, it is unclear whether similar patterns would emerge for individuals involved in LGBTQ+ relationships or individuals from other cultures. As a result, it would be informative to extend this research to a broader and more diverse group of participants. The third limitation is that we relied on self-report data, so it is possible that our respondents may have underreported some of the more negative aspects of their feelings or behaviors.

The final limitation is that we focused on subclinical levels of BPF, so it is possible that our results may not generalize to individuals who meet the diagnostic threshold

for BPD. Other features of our samples suggest that they may not be representative of the larger population. Although we sampled from the community, Prolific participants may have characteristics that distinguish them from individuals that do not have Prolific user accounts. We also sampled only from heterosexual couples and those that were in relationships lasting at least six months. Because individuals with BPF tend to experience more frequent break-ups and fewer long-term committed relationships (e.g., marriages) [25], many individuals with especially high levels of BPF may not have participated. In Study 2, participants were eligible only if their partners were also willing to participate. It is likely that this requirement—which is necessary to obtain dyadic data—may have discouraged couples from less healthy relationships from participating. Despite these limitations, we observed the predicted pattern of results; however, it is possible that even stronger and clearer patterns would be observed in a more representative sample.

Future research should continue to examine the connections that BPF have with romantic functioning such as relationship satisfaction and commitment. The current studies were focused on the use of MRB in response to jealousy. However, we did not explore specific behaviors despite some evidence that men and women engage in different types of mate retention [30,31]. Furthermore, other behavioral outcomes in response to jealousy would also be worth exploring, such as self-injurious behaviors, substance abuse, consideration of infidelity, or intentions to terminate the relationship. It may also be beneficial to consider whether suspicious jealousy plays an important role in other relationships (e.g., familial relationships, friendships) for those with high levels of BPF.

Lastly, given the role that social cognitive biases play in the response to threat for individuals with BPF [38], future work should focus on examining the development of these impairments in individuals with high levels of BPF. Although not considered diagnostic of BPD, there is growing evidence that BPF are associated with deficits in theory of mind [67], which may be particularly pronounced when individuals with BPF are emotionally aroused [42,68]. These deficits may factor into the association between BPF and suspicious jealousy observed in the present studies.

## 5. Conclusion

Given the relatively high rates of intimate partner violence and other serious and destructive behaviors in individuals with BPF and their general struggles with interpersonal relationships [25], any insights regarding the feelings that precipitate these destructive tendencies is valuable and may be useful in treatment programs [34,69]. We have taken a novel approach to furthering the understanding of BPF in romantic relationships. We have shown that BPF are associated with suspicious jealousy and that this jealousy may mediate the associations that BPF have with both benefit-provisioning and cost-inflicting behaviors in romantic relationships. Suspicious jealousy is often problematic for romantic relationships and may stem from a poor understanding of others' mental states and motives that is compounded by a tendency to misattribute malevolent intentions to others, which characterizes individuals high in BPF.

**Author Contributions:** Conceptualization, V.Z.-H.; methodology, V.Z.-H.; formal analysis, V.Z.-H. and J.V.; data curation, V.Z.-H.; writing—original draft preparation, V.Z.-H.; writing—review and editing, J.V. All authors have read and agreed to the published version of the manuscript.

**Funding:** This research received no external funding.

**Institutional Review Board Statement:** These studies were conducted in accordance with the Declaration of Helsinki and approved by the Institutional Review Board of Oakland University (IRB Protocol #IRB-FY2022-380 approved on 15 June 2022).

**Informed Consent Statement:** Informed consent was obtained from all subjects involved in the study.

**Data Availability Statement:** The data presented in these studies are openly available on the Open Science Framework (accessed on 15 August 2023) at https://osf.io/kp784/.

**Conflicts of Interest:** The authors declare no conflict of interest.

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
