# Peer review of "Borderline Personality Features and Mate Retention Behaviors: The Mediating Roles of Suspicious and Reactive Jealousy"

_sexes, doi:10.3390/sexes4040033_

Round 1

Reviewer 1 Report

I appreciate the opportunity to review the manuscript titled “Borderline Personality Features and Mate retention Behaviors: The Mediating Roles of Suspicious and Reactive Jealousy.” Overall, it is a very well-written and thoughtful manuscript, with very clean and well-presented figures. I have three overall concerns.

 1. The introduction didn’t really mention or set up for any of the potential sex differences that are part of the results. For instance, in study 1, I was surprised when the authors revealed that they tested gender as a moderator. Similarly, it was unclear whether the authors expected to find sex differences, especially in study 2 with the dyads. Perhaps they could expand the intro slightly to include more of a rationale for this part of the study, and include any specific hypotheses they had (or if this was exploratory, stating that explicitly).

 2. For Figure 1, were the error terms for the two outcome variables correlated? Also, were the error terms for the two types of jealousy correlated? These weren’t included in the model, so I wasn’t sure. Also, in Figure 2, the authors do include these, but I noticed that some of them were missing, like the correlation between the error terms for female suspicious jealousy and male reaction jealousy, the correlation between female reactive jealousy and male suspicious jealousy, and the correlations between the two female jealousies and the two male jealousies. I wasn’t sure whether this was because they were not significantly correlated or something, so I wanted to make sure this was intentional, as this may impact the results. I would try to make sure this is clearer to the readers about why these were included/excluded.

 3. One other limitation of the current study was that it was conducted with individuals in heterosexual relationships only. Thus, it is unclear whether these findings would generalize to homosexual romantic relationships, especially given the potential sex differences found. 

Reviewer 2 Report

This cross-sectional study of community participants examined the model that high borderline personality features may have an indirect effect on mate retention behaviors mediated by suspicious jealousy. Both the findings from the monadic and dyadic sample studies supported this hypothesis and this research provides information about potential mechanisms that lead to interpersonal instability for persons with borderline personality features. The research is well presented but the authors should address the following:

1. On page 2, line 54, the abbreviation BPF is introduced but should be defined when introduced.

2. On page 10, line 392, the authors discussed that stable relationships may mitigate the problematic interpersonal behaviors associated with BPF. In the current research, does the length of the relationship moderate the association between BPF, suspicious jealousy and mate retention behaviors?

3. On page 10, line 429, the statement that indicated that BPD is resistant to treatment is inaccurate; at least 6 evidence-based psychotherapies existed that have been proven effective for suicide-related outcomes and BPD features in patients with BPD. This statement needs to be corrected because it perpetuates the stigma towards patients with BPD.

Round 2

Reviewer 1 Report

I reviewed the first version of this manuscript and believe the authors have adequately addressed all concerns raised by both reviewers. I have no further comments.